# Influencing Factors in the Synthesis of Photoactive Nanocomposites of ZnO/SiO_2_-Porous Heterostructures from Montmorillonite and the Study for Methyl Violet Photodegradation

**DOI:** 10.3390/nano11123427

**Published:** 2021-12-17

**Authors:** Is Fatimah, Gani Purwiandono, Putwi Widya Citradewi, Suresh Sagadevan, Won-Chun Oh, Ruey-an Doong

**Affiliations:** 1Department of Chemistry, Faculty of Mathematics and Natural Sciences, Universitas Islam Indonesia, Kampus Terpadu UII, Jl. Kaliurang Km 14, Sleman, Yogyakarta 55584, Indonesia; gani_purwiandono@uii.ac.id (G.P.); 20923014@students.uii.ac.id (P.W.C.); 2Nanotechnology & Catalysis Research Centre, University of Malaya, Kuala Lumpur 50603, Malaysia; drsureshnano@gmail.com; 3Department of Advanced Materials Science and Engineering, Hanseo University, Seosan-si 356-706, Chungnam, Korea; 4Institute of Analytical and Environmental Sciences, National Tsing Hua University, 101, Sec. 2, Kuang Fu Road, Hsinchu 30013, Taiwan; radoong@mx.nthu.edu.tw

**Keywords:** advanced oxidation process, photocatalyst, porous clay heterostructure, photodegradation

## Abstract

In this work, photoactive nanocomposites of ZnO/SiO_2_ porous heterostructures (PCHs) were prepared from montmorillonite clay. The effects of preparation methods and Zn content on the physicochemical features and photocatalytic properties were investigated. Briefly, a comparison of the use of hydrothermal and microwave-assisted methods was done. The Zn content was varied between 5 and 15 wt% and the characteristics of the nanomaterials were also examined. The physical and chemical properties of the materials were characterized using X-ray diffraction, diffuse-reflectance UV-Vis, X-ray photoelectron spectroscopy, and gas sorption analyses. The morphology of the synthesized materials was characterized through scanning electron microscopy and transmission electron microscopy. The photocatalytic performance of the prepared materials was quantified through the photocatalytic degradation of methyl violet (MV) under irradiation with UV and visible light. It was found that PCHs exhibit greatly improved physicochemical characteristics as photocatalysts, resulting in boosting photocatalytic activity for the degradation of MV. It was found that varied synthesis methods and Zn content strongly affected the specific surface area, pore distribution, and band gap energy of PCHs. In addition, the band gap energy was found to govern the photoactivity. Additionally, the surface parameters of the PCHs were found to contribute to the degradation mechanism. It was found that the prepared PCHs demonstrated excellent photocatalytic activity and reusability, as seen in the high degradation efficiency attained at high concentrations. No significant changes in activity were seen until five cycles of photodegradation were done.

## 1. Introduction

Photocatalytic degradation of organic compound-containing wastewater is one of the important schemes in advanced oxidation processes (AOPs). Photocatalytic oxidation utilizing heterogeneous catalysts has received great attention for processing wastewater during the last decade due to its advantages compared to previous popular methods such as adsorption, chemical oxidation, and ozonation [1,2,3]. The process, which includes oxidation accelerated by solids, can produce complete oxidation in a continuous process with simpler handling and no need for further treatment of the material. Some metals and metal oxides have been shown to be efficient photocatalysts for water and wastewater treatments [4,5,6]. In addition, the enhancement of photocatalytic activity, mainly related to their chemical and environmental stability, can be achieved by using solid catalysts to support metal and metal oxide nanoparticles homogeneously [7]. Various solid supports have been reported to enhance the performance of metal and metal oxide catalysts for AOPs, and undoubtedly, the utilization of clay as a support material has received intensive interest [8,9,10,11]. Previous works reported the enhanced performance of clay-supported metal oxides in AOPs, which was mainly related to their stability in hostile environments and their recyclability and reusability. Such clay-supported metal oxides of ZnO, ZrO_2_, SnO_2_, and TiO_2_-pillared clays were characterized and noted to have distinctive photocatalytic activities, with the supportive adsorptive capability being attributed to the porosity of the material [9,12,13,14]. Based on some considerations, such as the range of band gap energy, chemical stability, and cost effectiveness, ZnO was chosen as a good alternative to TiO_2_ as a photocatalyst. Besides the formation of pillared clay, ZnO/clay nanocomposites were also reported in porous clay heterostructures. With the different schemes of synthesis, the porous clay heterostructure of ZnO/SiO_2_ showed a significant increase in specific surface area, which is advantageous for the adsorption mechanism involved in photocatalytic degradation [15,16].

As in many other studies on nanocomposite preparations, the structural and morphological properties of the nanocomposites depend on the synthesis method, which include solvothermal, coprecipitation, hydrothermal, and microwave-assisted syntheses. The use of these different methods results in varying characteristics of metal oxide/clay nanocomposites [17,18,19]. Previous studies reported the time-efficient synthesis of ZnO/SiO_2_ porous clay heterostructures (ZnO_2_/SiO_2_-PCH) from montmorillonite using the microwave-assisted method, which was then compared to the synthesis using the hydrothermal-assisted method [20,21,22]. An extremely high specific surface area compared to that of the parent clay was reported, with a higher specific surface area being obtained through the microwave-assisted method. However, the effect of the ZnO_2_/SiO_2_-PCH synthesis method on its photoactivity has not yet been studied. Moreover, a more intensive study on the effect of the amount of Zn, which serves as the photoactive component, on the physicochemical characteristics of ZnO_2_/SiO_2_-PCH is required for optimization. In many studies of zinc oxide-immobilized nanocomposite, the range of Zn content was set at 5–15 wt% [23,24].

Taking these into consideration, the present work aims to synthesize ZnO_2_/SiO_2_-PCH nanocomposites using natural Na montmorillonite. This study aims to compare the hydrothermal-assisted method and microwave-assisted method in the synthesis and investigate the effect of Zn content on the physicochemical and photocatalytic properties of nanomaterials. The following characterization techniques were employed to study the physicochemical properties of materials: X-ray diffraction (XRD), scanning electron microscopy with energy-dispersive X-ray spectroscopy (SEM-EDX), transmission electron microscopy (TEM), UV-visible diffuse reflectance spectroscopy (UV-DRS), and X-ray photoelectron spectroscopy (XPS). The photoactivity was evaluated through the photocatalytic oxidation of methyl violet. The photocatalytic activity was also studied by comparing the use of UV light and visible light, as well as the addition of scavengers in the photocatalytic system.

## 2. Materials and Methods

### 2.1. Materials

Montmorillonite (Mt) obtained from Sidoarjo, East Java, Indonesia, was acid-activated prior to use. The activation was done by refluxing the 20% *w/v* clay suspension in 1 N H_2_SO_4_ for 4 h, followed by washing with H_2_O. Zinc acetate, tetraethyl orthosilicate (TEOS), cetyltrimethylammonium bromide (CTMA), tetramethylammonium hydroxide (TMAOH), and methyl violet (MV) were purchased from Merck-Millipore (Darmstadt, Germany).

### 2.2. Preparation of ZnO_2_/SiO_2_-PCH 

Nanomaterials of ZnO_2_/SiO_2_-PCH were prepared by mixing 5 g of Mt with 3 g of CTMA bromide in 300 mL of aquadest followed by stirring overnight. Next, 1 g of TMAOH was added to the suspension, followed by stirring for 4 h. The pillaring agent was prepared by mixing zinc acetate and TEOS (at a Zn:Si molar ratio of 1:4) in 5 mL of isopropanol under vigorous stirring for 2 h. The pillaring agent was added to the CTMA–Mt suspension with a varied Zn content of 5, 10, and 15 wt%. The obtained mixtures were then subjected to two synthesis methods: The hydrothermal-assisted method (HT) and the microwave-assisted (MW) method. For the microwave-assisted method, the mixture was microwave-irradiated for 30 min, while for the hydrothermal method, the mixture was kept in an autoclave at 150 °C for 3 days to achieve gel formation. The 3 day treatment was set with reference to previous time optimization for the hydrothermal process and based on previous study of some PCH syntheses [25,26]. The formation of porosity will optimally occur at 1–3 days. The gels obtained from each procedure were dried in an oven at 60 °C overnight before sintering at 500 °C for 2 h. The ZnO_2_/SiO_2_-PCH samples synthesized using the HT method were encoded as PCH-5/HT, PCH-10/HT and PCH-15/HT for the samples containing 5, 10, and 15 wt% Zn, respectively. A similar naming convention was used for the samples synthesized using the MT method. Figure 1 displays a schematic representation of the Zn/Si-PCH synthesis.

### 2.3. Physicochemical Characterization

As-prepared ZnO_2_/SiO_2_-PCH materials were analyzed using X-ray diffraction (XRD) on a Shimadzu X6000 instrument (Kyoto, Japan). Ni-filtered Cu-Kα was utilized as radiation source and the scan rate was 2°/min. Morphology of the samples were recorded using scanning electron microscope (SEM)-Phenom X (Waltham, MA, USA), and the high-resolution transmission electron micrographs were obtained on a JEM 2010 (HR) transmission electron microscope (TEM) (Pleasanton, CA, USA). The samples were pressed and degassed before being deposited onto a copper mesh grid. Nitrogen adsorption/desorption isotherms of the samples were measured at −77 K°C using a Quantachrome instrument (Boynton Beach, FL, USA). Degassing at 90 °C for 10 h was performed prior to each analysis, and based on the adsorption/desorption data, specific surface areas were calculated based on the Brunauer–Emmett–Teller model (BET specific surface area). Other parameters of pore distribution, pore volume, external surface area were calculated based on the Barrett, Joyner, Halenda (BJH) method and t-method, respectively. Elemental analyses of samples were measured by X-ray fluorescence (XRF) PANalytical Minipad 4(Malvern Panalytical, Malvern, UK). Optical properties of materials consist of diffuse-reflectance spectroscopy analysis (UV-DRS) and photoluminescence (PL) spectra were recorded on JASCO V-760 instrument (Jasco Inc., Easton, MD, USA). X-ray photoelectron spectroscopy (XPS) analysis was carried out by using a Thermo Scientific Escalab 250 instrument (Thermofisher, Waltham, MA, USA).

### 2.4. Photocatalytic Activity Measurement

The photocatalytic activity of the samples was examined through the photocatalytic degradation of MV under UV and visible light irradiations. For each test, about 250 mL MV aqueous solution with a concentration of 20 mg/L was mixed with 0.2 g photocatalysts and 2% H_2_O_2_ solution in a water-jacketed Pyrex vessel equipped with a UV lamp in the center. The UV light source was a 20 watt-Philips lamp with an intensity of 39.99 MW/Cm^2^; meanwhile, a Philip Xenon lamp 20 watt with an intensity of 31.22 MW/Cm^2^ was utilized as the visible light source. The wavelengths of the UV lamp and the xenon lamp were 254 nm and 485 nm, respectively. The photoreactor was water-cooled to a stable temperature of 25 °C to avoid heating effects caused by irradiation. Before the UV illumination, the mixture was stirred in dark conditions to achieve adsorption–desorption equilibrium. Sampling was conducted through the sequential pipetting of the solution from the photoreactor. The concentration of MV in the aliquots was determined through the colorimetric method using a HITACHI U-2010 spectrophotometer (Hitachi Asia Ltd, Singapore). The adsorption experiments were also performed to study the adsorption mechanism in the absence of UV light irradiation was studied as well using the same set-up, excluding the UV lamp. The study on the recyclability of the photocatalyst was conducted by first filtering the samples and washing them with water. The spent photocatalysts were dried at 100 °C overnight before repeating the photocatalytic activity measurements.

The degradation efficiency of the photocatalysis was calculated based on the following Equation (1):(1)$$Degradation\;efficiency\;\left(DE\right)\left(\%\right)=100\;\times \;\left(\frac{C_{0}-C_{t}}{C_{0}}\right)$$
where *C*_0_ and *C_t_* are initial concentration and concentration of MV at time *t*.

## 3. Results

### 3.1. Structural Analysis of Materials

The XRD patterns of the prepared materials in comparison with that of Mt as raw material are presented in Figure 2.

The XRD pattern of the Mt raw material shows the diffraction peak d_001_ at 6.38° that can be attributed to the layered structure and corresponds to a basal spacing of 1.09 nm. Additionally, two-dimensional diffractions at 19.75° and 35.04° are present, which can be assigned to the (211) and (004) reflections, respectively [18,27,28]. The XRD patterns of all the PCH samples show insignificant differences when compared to the Mt patterns, revealing that the montmorillonite structure is still maintained. Another peak positioned at 28.5° corresponds to the (111) planes of cristobalite, and the peak at 26.7° is an indication of quartz. Both of these phases are usually found in natural minerals [29]. By using small angle identification, the PCH samples show the distinctive d_001_ reflection shifting to lower values of 2θ. It is shown that the increasing d_001_ spacings are about 1.51 nm to 3.17 nm, which demonstrates the effect of Zn content and synthesis methods. In general, the MW method resulted in larger increases in the spacings compared to the HT method, and for both methods, the highest interlayer space was found in the 10 wt% Zn samples. In addition, there were no peaks that represented ZnO in any of the samples. The absence of a ZnO peak implies that the ZnO in the PCH structure is not in the aggregate form. The dispersion of ZnO together with SiO_2_ tends to give ZnO in the nanoparticle form and results in homogeneous distribution of ZnO. In particular, PCH-10/MW shows an additional peak corresponding to a spacing of 1.51 nm, reflecting partial delamination in the formation of paraffin monolayers [30,31].

The availability of ZnO is further examined by SEM-EDX analysis, and the results are presented in Figure 3 and Table 1. The morphology of samples represents thistle-like structures along with the flaky structures of PCH samples, which are characteristics of the two-dimensional ZnO nanoparticle surface [32,33,34]. The elemental analysis based on EDX spectra indicated that the Zn percentages are slightly higher compared to the targeted and calculated amounts, which respect the mass loss of some impurities such as sodium, aluminium, and other minerals during preparation steps. In addition, the increasing Si content in all PCH samples was due to the addition of silica from TEOS (Table 1).

The N_2_ adsorption isotherm presented in Figure 4 depicts the increasing adsorption capability of the montmorillonite after modification. All samples exhibit the type IV adsorption isotherm showing typical H3-type hysteresis, which reveals the layered structure-based materials. The pore distribution suggests there is a predominance of mesopores (2–50 nm), which were significantly increased in PCHs and were considerably abundant in PCH15/MW. Significant hysteresis loops are reflected by the mesopore modus at 22 nm. Furthermore, using data from the adsorption/desorption isotherm, microtextural parameters of the BET surface area, pore volume, and external surface area of the materials were obtained and are presented in Table 2. The data indicate the effect of modification on the development of mesoporous structures for higher adsorption capability [35]. At a Zn content of 5 wt%, it is seen that the MW method resulted in higher BET specific surface area compared to the HT method, but as the Zn content was increased to 10 and 15 wt%, the HT method produced higher values. Additionally, a decrease in BET specific surface area with the increase of Zn content was observed in both methods, which could be attributed to pore blocking due to the irregular porous structures. This assumption is also proven by the analogue trend of the external surface area and micropore area of the samples. As the BET specific surface area decreased, both parameters decreased. A similar pattern related to the effect of metal oxide content on surface adsorption capability is also observed in the modification of montmorillonite using Fe_3_O_4_ and SnO_2_ [36]. The correlation between the detected interlayer space d_001_, pore size distribution, BET specific surface area, external surface area, and pore volume of PCH10/MW represents the formation of a non-homogeneous pillar structure, generally called “house of cards structure”. A non-uniform formation of mesopores is the main reason for the lower specific surface area and external surface area. This also reflects the less ordered distributed ZnO/SiO_2_ pillars obtained from the MW method compared with the HT method.

TEM analysis was conducted to confirm the formation of porous structures. The selected and distinctive images are depicted in Figure 5. The higher basal spacing d_001_ of PCH is seen in the images, and moreover, the homogeneously distributed SiO_2_ is confirmed. A regular particle size represented the mesoporous silica in the materials; meanwhile, the inter-layer distances of approximately 2.2–3.1 nm indicate an increasing interlayer region (Figure 5a–d). In particular, the TEM images of PCH15/MW and PCH15/HT exhibit the identified house of cards structure as the mesopores sized at around 20–60 nm, confirming the presence of irregular conformations at higher ZnO_2_/SiO_2_ content. Similar patterns were also found in other porous clay heterostructures using silica and titania-silica [37,38]. The schematic representation of the structure can be seen in Figure 6. The formation of zinc-oxide pillars in the PCHs was initiated by the intercalation of CTMA into the interlayer region of silica-alumina sheets. Furthermore, the insertion of silica-zinc precursor replaces the exchangeable cations that are the native cations in clay sheets. Irregular distribution of the silica-zinc oxide polyoxocations contributed to the creation of mesopores as a result of the irregular conformation of the sheets.

### 3.2. Optical Properties of Materials

The optical properties of photocatalysts are the crucial parameters for photocatalysis systems; therefore, the prepared PCHs were analyzed using UV-DRS and photoluminescence spectrum measurements. Some UV-DRS spectra and Tauc plots of the samples are presented in Figure 7, in which the band gap energy values are calculated by plotting energy (hν) versus (αhν)^2^ curves based on the Kubelka–Munk theory. Based on the plots and details of the data presented in Table 3, it is conclusively found that, in general, higher Zn content resulted in higher band gap energies. Band gap energy values within the range of 3.25–3.33 eV were obtained from ZnO/SiO_2_-PCH samples. In particular, a band gap energy value of 3.33 eV was obtained from the 15 wt% Zn samples from both methods. The data highlight the contribution of ZnO as a photoactive material to the optical properties, which is not notifiable correlated with the particle size, as a result of the quantum size effect. The contribution of crystallite defects and oxygen vacancies caused by the ZnO distribution is more significant than the effect of the particle size. This suggests a similar phenomenon to that recorded in ZnO nanocomposites such as ZnO/SiO_2_ [40,41]. The effect of Zn content on optical features is confirmed by the PL spectra presented in Figure 7c. It is observed that the PL spectra of PCH10/HT and PCH15/HT show a similar pattern of excitation spectra, which lay in the 200–600 nm wavelength range with intense peaks in the 350–450 nm and 525–575 nm regions.

In all ranges, PCH10/HT exhibited less intensity compared to PCH15/HT, which proportionally correlated with the quantitative ZnO amount as photoactive particles. This underscores the capability of PCHs to interact with photons within the UV and visible regions.

### 3.3. Photocatalytic Activity

The photocatalytic activity of the samples was tested by evaluating the kinetics of MV photocatalytic degradation under the illumination of UV and visible lamps. To ensure the reliability of the experiments, photocatalytic experiments use a control setup without any photocatalyst and oxidant, and a setup without photocatalysts is also conducted. Figure 8 shows the kinetics of MV photodegradation using various photocatalysts. For both light sources, it can be seen that there is no measurable photodegradation in the absence of photocatalysts and in the absence of both oxidants and photocatalysts until 2 h of treatment. This indicates that even though the reaction occurs thermodynamically, the reaction rates are slow and not significantly detected during the time of treatment. Furthermore, the photoactivity of PCH samples is clearly seen, the rate of photodegradation increased in proportion with the decrease of MV concentration. Due to similar plots of MV degradation by Mt under the different light sources, it can be concluded that decrease in MV concentration was caused by the adsorption mechanism. To ascertain the existence of degradation mechanisms, UV-Visible spectrophotometry analyses were utilized to compare the spectral change of treated solutions between Mt and PCH15/HT, and the results are depicted in Figure 9.

With the increasing time of treatment, the adsorption produces reduced absorbance spectra without any shifting of peaks. In contrast, the photocatalysis treatment resulted not only in reduced absorbance, but also in a blue shift from the characteristic peak at 586 nm. Typical blue shift is characteristic of the demethylation that was also investigated in photocatalytic degradation of MV by hematite/TUD-1 [42], MV by TiO_2_/Pd [42], methyl orange by CdS [43], and crystal violet by BaWO_4_ nanorods [44].

The photodegradation rates varied with each photocatalyst, but, in general, the faster reaction occurred under irradiation with UV light rather than with visible light. Under UV light, the degradation efficiencies (DEs) reached more than 90% at 30 min over all PCH samples, while DEs of only 59% were achieved under visible light illumination. This indicates that the photocatalytic mechanism exhibited by ZnO is a result of electron excitation under UV light and is faster compared to the degradation under visible light.

The study on kinetics data revealed the fitness of MV photodegradation with pseudo-second-order kinetics in Equation (2):(2)$$\frac{1}{C_{t}}=k_{obs}t+\frac{1}{C_{0}}$$
where *C*_0_ is the initial concentration of MV, *C_t_* is the concentration of MV at time *t*, and *k_obs_* is the observed kinetics constant. Some of the pseudo-second-order kinetics plots are presented in Figure 10, and the calculated kinetics parameters and equations are presented in Table 3.

From the k_obs_ and DE values, it was obtained that PCH15/HT was the most photoactive material, where an almost complete photodegradation of MV after 2 h was observed, which is potentially related to the band gap energy and Zn content of the nanocomposite. In general, the data indicate that the band gap energy governs the photoactivity and that surface parameters such as BET specific surface area contribute to the degradation mechanism.

More detailed analyses of the DEs of UV light-assisted photodegradations revealed the significant role of the band gap energy in the photocatalytic mechanism. The statistical calculation reflects the correlation coefficient of band gap energy and DE is 0.985. Generally speaking, from the photocatalysis under UV light illumination, it was obtained that the higher Zn content gave the higher DE values as the more photoactive site caused more reduction-oxidation to occur. In addition, the higher specific surface area contributes to enhancing the degradation via adsorption mechanism. A similar phenomenon on the role of specific surface area and external surface area was also reported in the photocatalysis by hydrothermally synthesized-titanium pillared clay [45], tin oxide-montmorillonite [21], and zinc oxide/bentonite [46]. However, the role of Zn content is not linearly correlated with the photocatalytic activity in visible light exposure. It is concluded that the photoactivity is derived from the combination between surface profile and the optical properties of the photocatalyst, which is particularly governed by the surface character of the photocatalyst. The comparison between two synthesis methods specifies that the hydrothermal-assisted method gave a better surface profile compared to the microwave-assisted method.

### 3.4. Effect of pH and Initial Concentration of MV

The effect of MV concentration on kinetics of photodegradation reaction was evaluated by measuring the initial rate of the reactions using PCH15/HT photocatalyst under UV and visible light. The data are depicted in Figure 11. It can be seen from the chart in Figure 11a that MV concentration contributes to the increasing reaction rate, which is consistent with the fitness of the kinetics to pseudo-second-order kinetics. The consistence effect of the light source is also found for all varied concentrations. In the perspective of DE in Figure 11b, it is noteworthy that the PCH samples prepared in this work exhibited excellent photocatalytic activity, as a DE of above 95% was achieved for a highly concentrated solution (50 mg/L). For comparison, a DE value of about 92% was achieved by using Pt/TiO_2_ photocatalysts at a dosage of 3 g/100 mL for a 5 mg/L MV solution, which is comparable to the PCHs, although the PCHs required a lower dose and a shorter time of treatment. The photocatalytic activities of the prepared samples in this work in comparison with other photocatalysts are presented in Table 4.

Furthermore, in order to study the effect of pH, photocatalysis reactions at various pH levels (pH = 4, 7, 9, and 11) were examined. The kinetics plots with the initial rate comparison are presented in Figure 11c and the bar chart of the initial rates are presented in Figure 11d.

The highest initial rate presented in the kinetics plots showed that a pH of 7 is optimal for photodegradation; meanwhile, the rate decreases in acidic and basic conditions. The data suggest that the surface charge amphoteric properties are involved in the mechanism, as they influence the interaction between the photocatalyst surface and the target molecule. In acidic conditions, the photocatalyst surface is covered with protons that suppress the affinity of the surface with MV due to the potential electrostatic repulsion. On the other hand, basic conditions are enriched with negative charges, and hydroxide ions will attract MV molecules through electrostatic interactions before the MV molecules can interact with the nanomaterial surface and oxidants produced from the photocatalytic mechanism. The decomposition of H_2_O_2_ is retarded by the abundance of either OH^-^ or H^+^, so the propagation steps occur at a lower rate [27,28]. A similar phenomenon is seen in the photocatalytic oxidation by Fe_2_O_3_ [55], TiO_2_ [29,30], and Fe_3_O_4_@SiO_2_@ZnO [56].

### 3.5. Effect of Scavengers

In order determine the reactive species that govern the photodegradation mechanism, the kinetics of photodegradation in the presence of scavengers were evaluated by the addition of isopropanol (IPR) and ascorbic acid (AA) as the hydroxyl radical and hole scavengers, respectively. The kinetics plots are depicted in Figure 12.

It is seen that the reaction rate decreased with the addition of IPR, which indicates the role of radicals in the mechanism. The existence of IPR in the solution has the capability to trap hydroxyl radicals and other radical forms produced from the interaction between ZnO and photons in the initiation step. The trapping of these radicals suppressed the propagation steps, which did not allow them to fulfill their role in further MV oxidation. In contrast, the addition of AA increased the reaction rate by blocking photogenerated holes, which resulted in the inhibition of electron-hole recombination and the easy migration of electrons on the nanomaterial surface. These allowed for the production of radicals through interactions with the solvent or O_2_. These results imply that both radicals and hole formation via photon interaction with photoactive ZnO play significant roles in governing the rate of reaction. A similar effect was also reported in research on the photocatalytic activities of Fe_2_O_3_ nanoparticles [57] and SnO_2_ [21].

### 3.6. Photocatalyst Reusability

The reusability of photocatalysts is one of the most required features for their applicability on an industrial scale. The DEs of the samples were measured over photodegradation cycles under UV and visible light. The data are expressed using a bar chart in Figure 13. It can be seen that, in general, DE values were maintained with no significant changes until the fifth cycle, as the DE values decreased by no more than 10% under both light sources. However, it is seen that the use of visible light resulted in better stability, which is mainly related to a milder oxidation-reduction mechanism that affects the photocatalyst surface less. In order to confirm this, the surface characteristics of PCH15/HT after five cycles of utilization were evaluated using XPS analysis, and the spectra are presented in Figure 13.

The XPS spectra of the fresh material indicated the existence of zinc oxide in the photocatalyst sample, as shown by the presence of Zn peaks (1126.50 eV) beside Si (103.10 eV), O (531.47 eV), and Na (1022 eV). The presence of Zn is identified by the doublet peaks (Figure 14b), which are ascribed to Zn 2p_3/2_ and 2p_1/2_ core levels, representing the presence of Zn^2+^ ions in an oxide environment. Survey scans of fresh and used materials after the fifth cycle show no significant change in composition, with similar Zn 2p peaks (Figure 14a). However, a slight change is seen in the deconvolution of O1s peaks (Figure 14c). The utilized material exhibits peaks at 529.1 eV, 530.9 eV, and 532.4 eV. The peaks at 529.1 eV and 530.9 eV can be ascribed to O^2-^ ions and Zn-O of the wurtzite structure, and the peak at 530.9 eV belongs to the Zn-OH bonding [15,58,59]. Zn-OH corresponds to the surface ZnO coordinated with H_2_O and the result of the dihydroxylation process in the synthesis. The change in the ratio of their intensities is an indication of electronic change after utilization in the oxidation-reduction mechanism of photocatalysis. This change is confirmed by the atomic ratios of Zn to O, as presented in Table 5. The Zn/O atomic ratio decreased from 0.531 in the fresh material to 0.518 after the fifth cycle. Generally speaking, the prepared PCHs exhibited excellent photocatalytic activity features.

## 4. Conclusions

A study on the preparation, characterization, and photocatalytic properties of ZnO/SiO_2_ porous heterostructures (PCHs) was successfully conducted. The physicochemical characteristics of PCHs were found to be influenced by the synthesis methods and the Zn content of the material. A significantly increased specific surface area of the material was achieved, which indicated the formation of the porous structure. The calculated band gap energies are within the range of 3.2–3.33 eV. Photocatalytic studies through MV photodegradation suggest that specific surface area and band gap energy significantly contribute to the high degradation efficiency achieved at high MV concentrations. The supporting clay contributes to the material reusability, as seen in the absence of any notable changes in the degradation efficiency until the fifth cycle of usage. From the varied synthesis method, it can be seen that the hydrothermal method produced higher specific surface area, which plays an important role in enhancing the photodegradation mechanism.

## 5. Patents

There is no patent resulting from the work reported in this manuscript.

## Figures and Tables

**Figure 1 nanomaterials-11-03427-f001:**
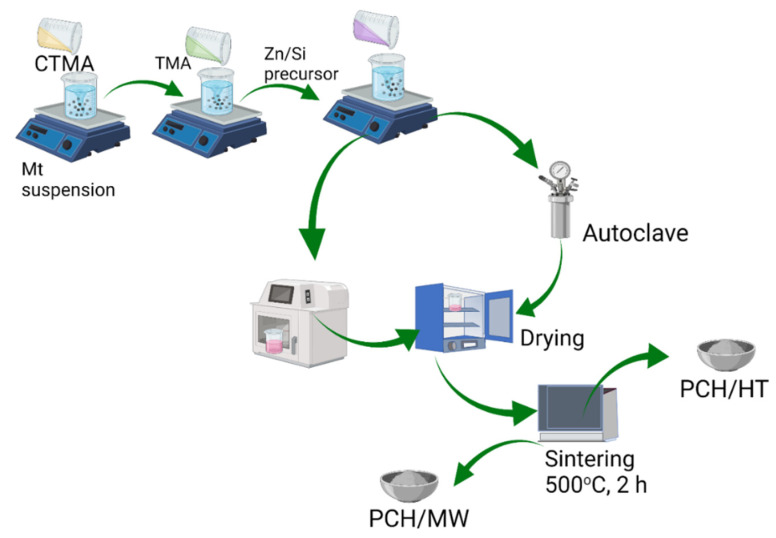
Schematic representation of ZnO_2_/SiO_2_-PCH synthesis.

**Figure 2 nanomaterials-11-03427-f002:**
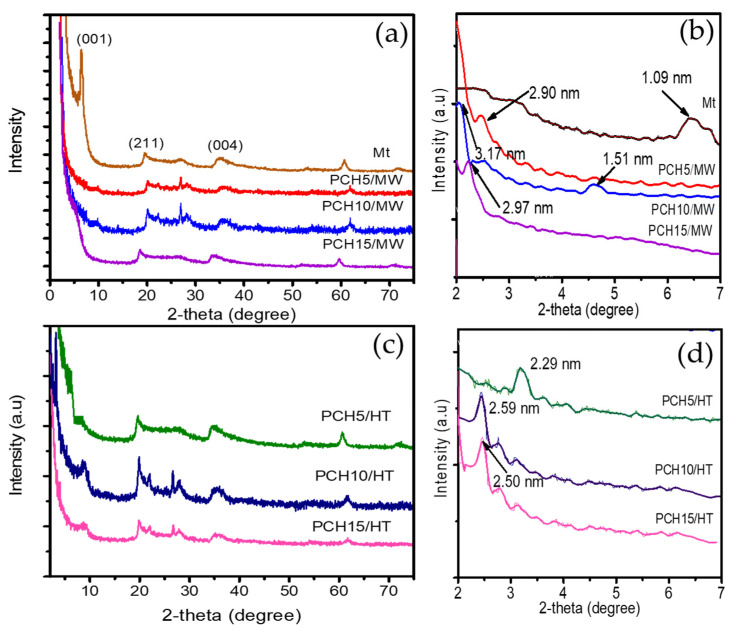
XRD and small angle XRD patterns of ZnO_2_/SiO_2_-PCH synthesized by (**a**,**b**) MW method (**c**,**d**) HT method.

**Figure 3 nanomaterials-11-03427-f003:**
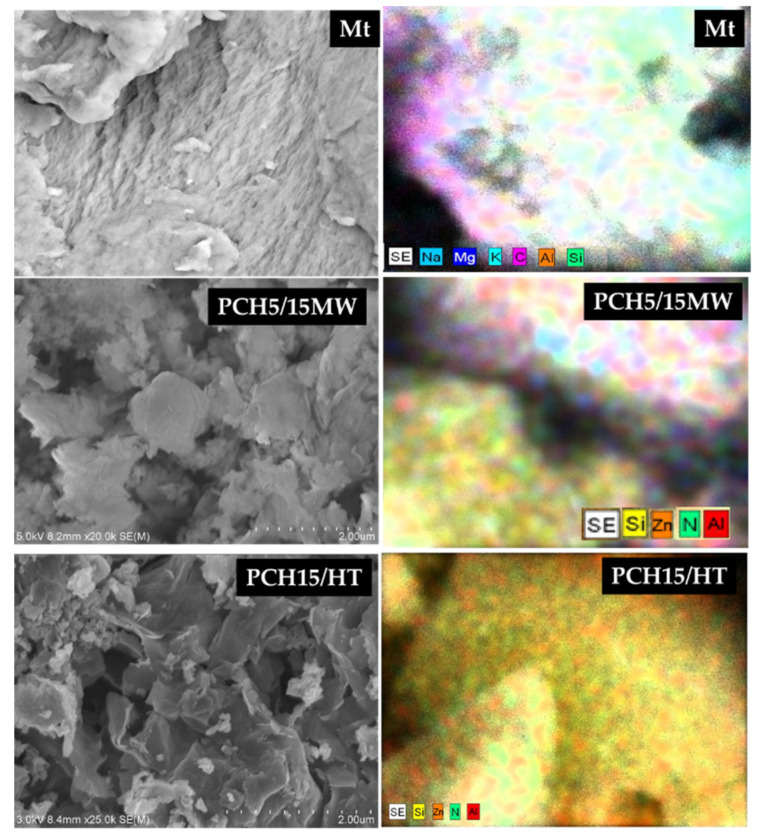
SEM images of ZnO_2_/SiO_2_-PCH samples.

**Figure 4 nanomaterials-11-03427-f004:**
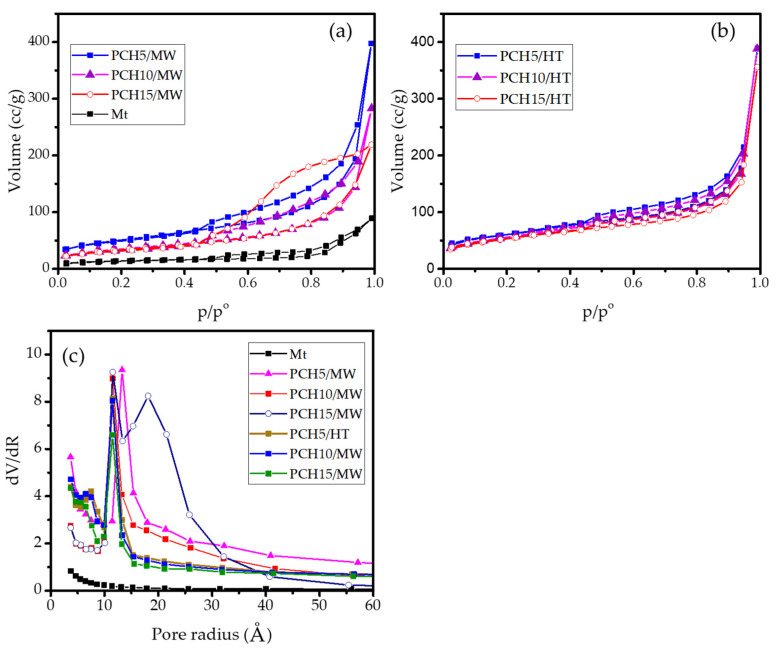
(**a**,**b**) Adsorption–desorption profile of materials; (**c**) Pore distribution of materials.

**Figure 5 nanomaterials-11-03427-f005:**
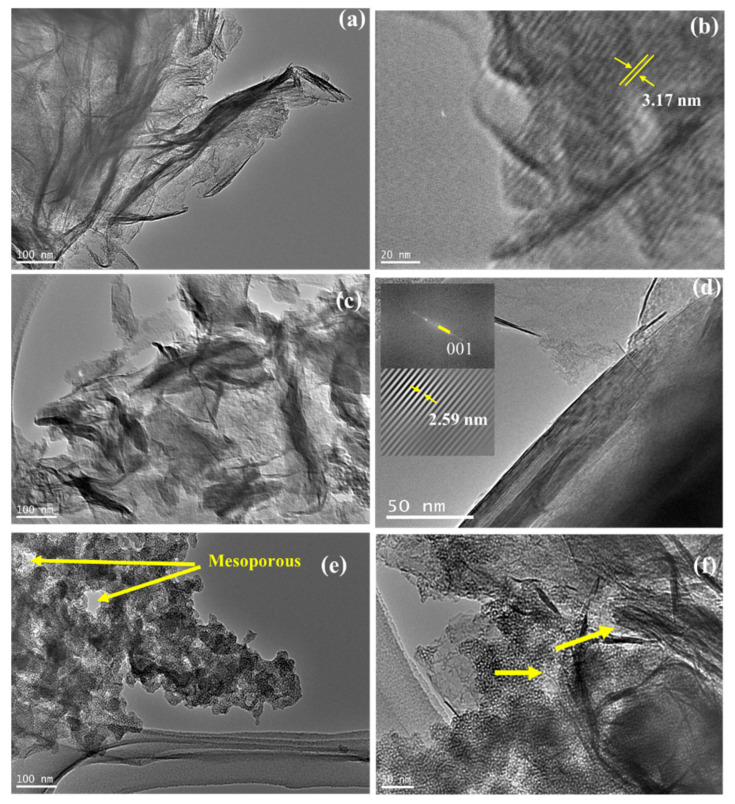
TEM images of: (**a**,**b**) PCH5/MW; (**c**,**d**) PCH10/HT; (**e**) PCH15/MW; (**f**) PCH15/HT.

**Figure 6 nanomaterials-11-03427-f006:**
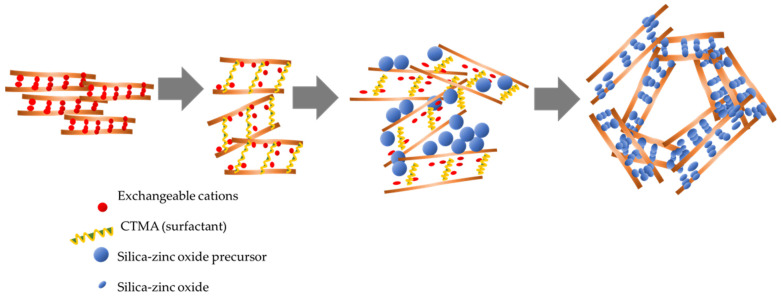
Schematic representation of house of cards formation [39].

**Figure 7 nanomaterials-11-03427-f007:**
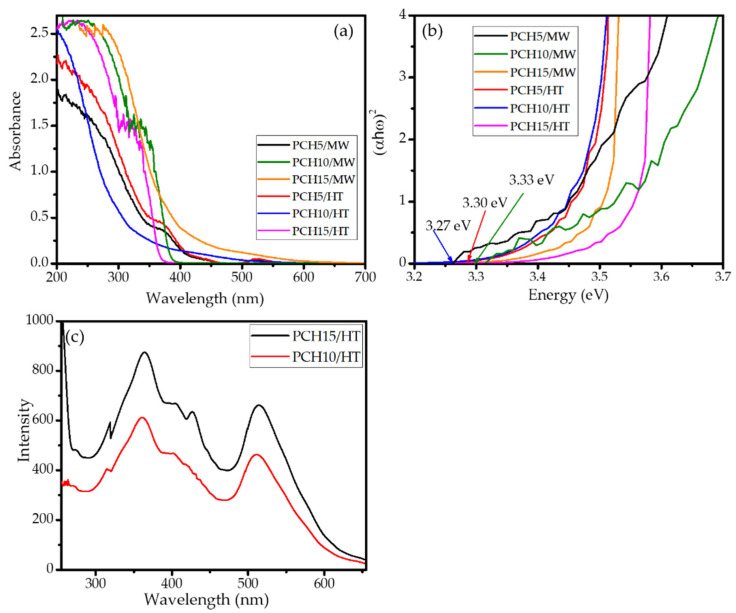
(**a**). Exemplary UV-DRS spectra of ZnO/SiO_2_-PCH samples; (**b**). Tauc plots of ZnO/SiO_2_-PCH samples (**c**) PL spectra of PCH10/HT in comparison with PCH15/HT.

**Figure 8 nanomaterials-11-03427-f008:**
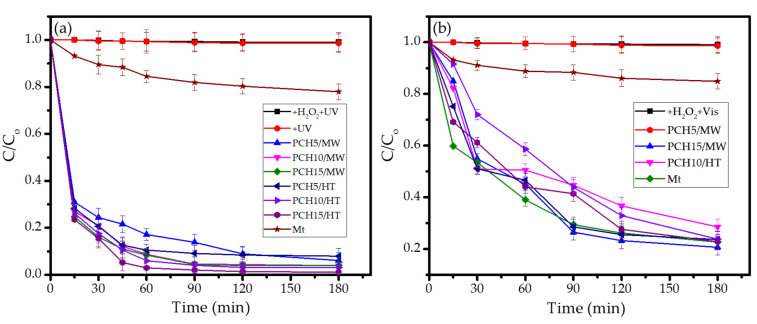
Kinetics of MV photodegradation over varied ZnO/SiO_2_-PCH samples in comparison with photolytic process and the treatment using Mt under (**a**) UV light; (**b**) visible light.

**Figure 9 nanomaterials-11-03427-f009:**
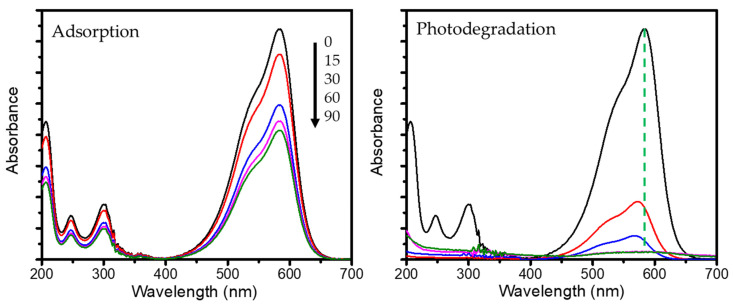
Comparison of UV-Visible spectra of treated solution by adsorption and photodegradation mechanism.

**Figure 10 nanomaterials-11-03427-f010:**
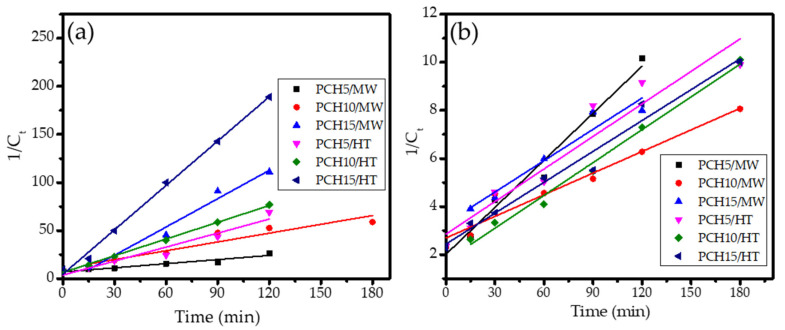
Pseudo-second-order plots of MV photodegradation over various ZnO/SiO_2_-PCH samples under (**a**) UV light; (**b**) visible light.

**Figure 11 nanomaterials-11-03427-f011:**
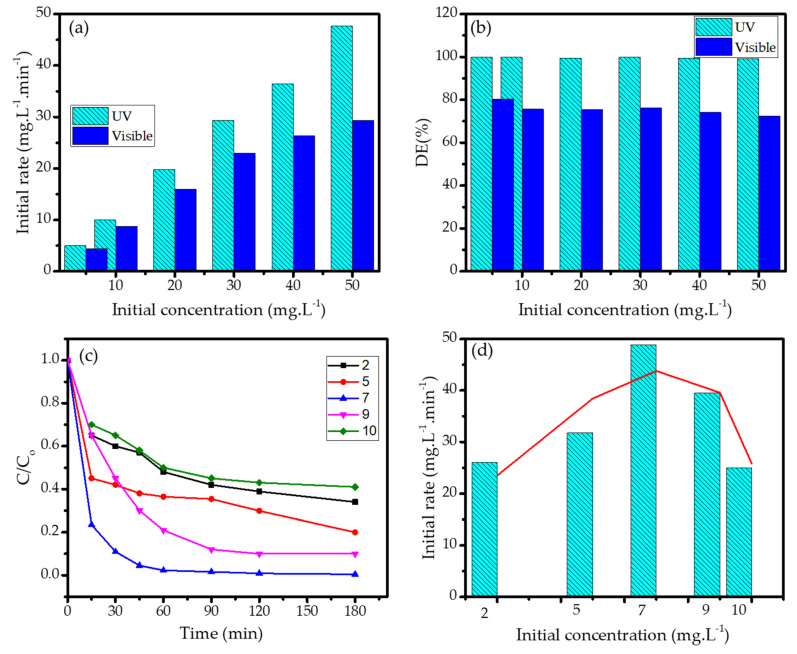
(**a**) Effect of initial concentration of MV on initial rate; (**b**) Effect of initial concentration of MV on DE; (**c**) Kinetics of MV photodegradation at varied pH; (**d**) DE at varied pH.

**Figure 12 nanomaterials-11-03427-f012:**
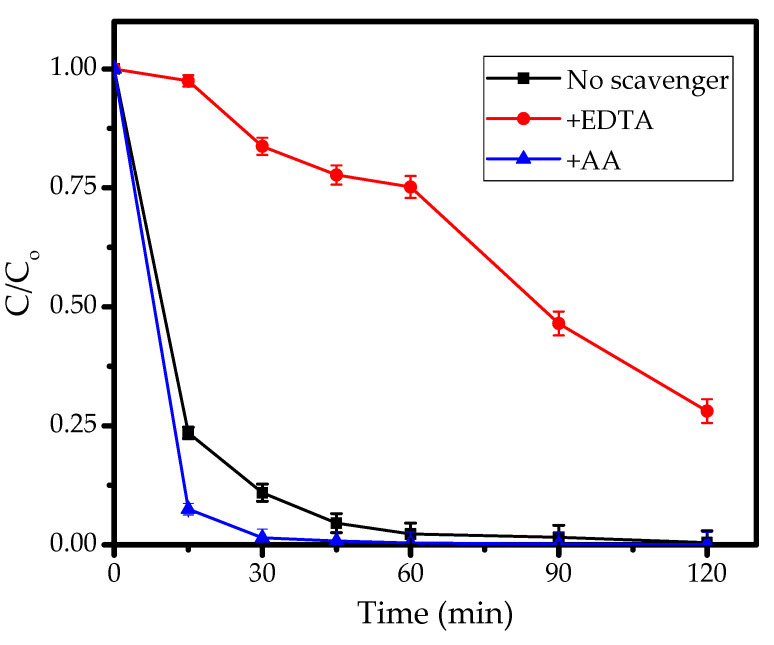
Kinetics of MV photodegradation in the presence and absence of scavenger.

**Figure 13 nanomaterials-11-03427-f013:**
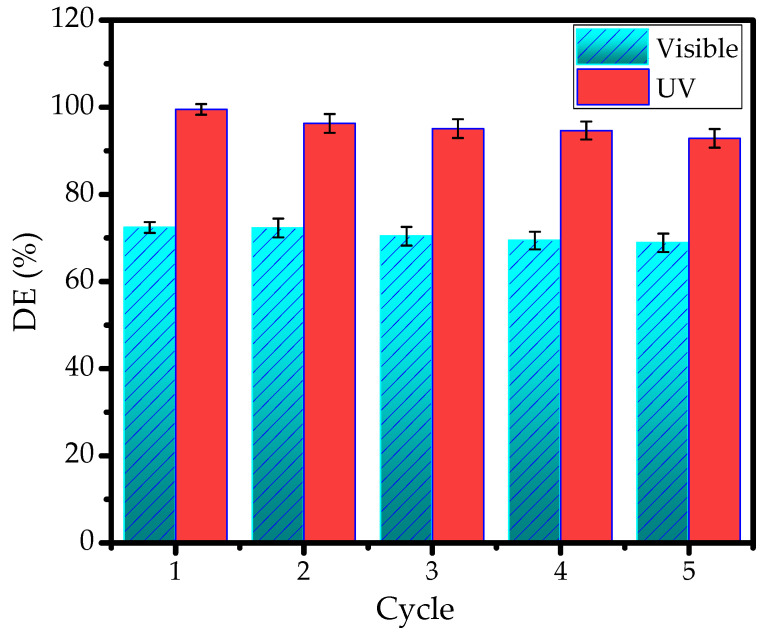
DE of MV photodegradation using PCH15/HT at first to fifth cycles.

**Figure 14 nanomaterials-11-03427-f014:**
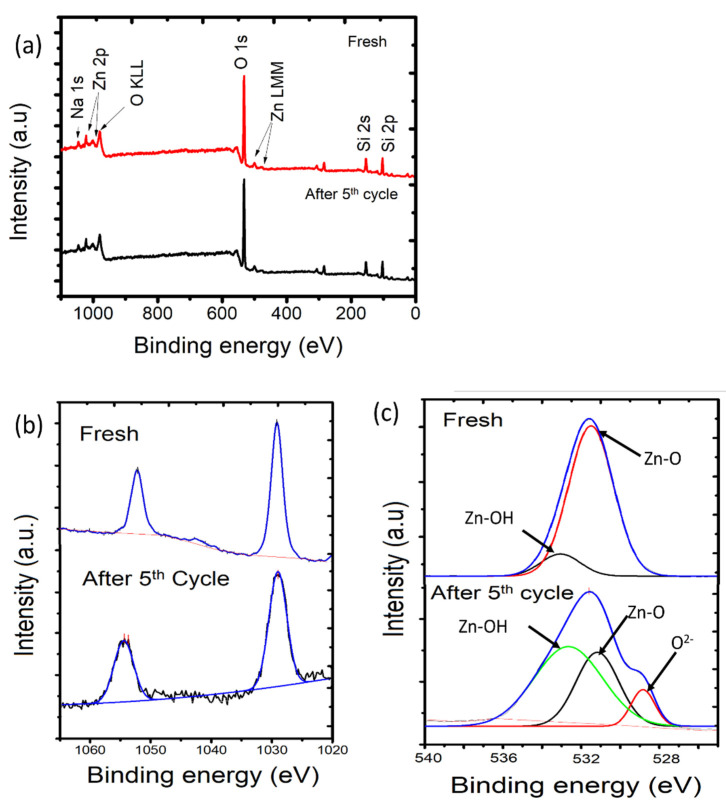
(**a**) Survey scan of PCH15/HT before and after use; (**b**) Spectra of 2p before and after use; (**c**) Deconvoluted O1s peaks before and after use.

**Table 1 nanomaterials-11-03427-t001:** Elemental analysis and d_001_ of ZnO_2_/SiO_2_-PCH materials in comparison with Mt.

Sample	Component (% at.)					d_001_ (nm)
	O	Si	Al	Na	Zn	
Mt	59.1	23.56	16.11	0.94	0	1.09
PCH5/MW	54.86	32.14	6.69	0.12	6.03	3.17
PCH10/MW	48.67	33.44	5.99	0.34	11.4	1.51; 3.17
PCH15/MW	45.22	31.12	6.06	0.61	16.61	2.97
PCH5/HT	49.15	36.76	7.24	0.47	5.83	2.29
PCH10/HT	46.81	34.09	8.27	0.04	10.77	2.59
PCH15/HT	45.89	30.69	7.23	0.07	15.98	2.50

**Table 2 nanomaterials-11-03427-t002:** Surface parameters and band gap energy of ZnO/SiO_2_-PCH samples.

Sample	BET Specific Surface Area (m^2^/g)	Pore Volume (cc/g)	External Surface Area	Band Gap Energy (eV)
Mt	68.90	0.023	10.8	-
PCH5/MW	706.40	0.295	269.62	3.27
PCH10/MW	561.57	0.284	213.65	3.30
PCH15/MW	538.03	0.237	105.63	3.33
PCH5/HT	687.26	0.269	155.38	3.29
PCH10/HT	663.42	0.278	141.43	3.29
PCH15/HT	648.51	0.218	119.77	3.33

**Table 3 nanomaterials-11-03427-t003:** Kinetics parameters and DE of MV photodegradation over various ZnO/SiO_2_-PCH sample.

Sample	Photon Source	Kinetics Equation	R^2^	k_obs_	DE at 120 min (%)
PCH5/MW	UV	1/C_t_ = 0.109 *t* + 7.88	0.983	0.109	91.08
PCH10/MW	UV	1/C_t_ = 0.424 *t* + 5.92	0.949	0.424	96.08
PCH15/MW	UV	1/C_t_ = 0.897 *t* + 1.32	0.988	0.897	98.60
PCH5/HT	UV	1/C_t_ = 0.376 *t* + 6.50	0.972	0.376	96.72
PCH10/HT	UV	1/C_t_ = 1.284 *t* + 6.85	0.996	1.284	96.88
PCH15/HT	UV	1/C_t_ = 1.614 *t* + 4.48	0.987	1.614	99.54
PCH5/MW	Visible	1/C_t_ = 0.064 *t* + 2.01	0.972	0.064	74.41
PCH10/MW	Visible	1/C_t_ = 0.027 *t* + 2.96	0.973	0.027	63.32
PCH15/MW	Visible	1/C_t_ = 0.043 *t* + 3.30	0.963	0.043	70.79
PCH5/HT	Visible	1/C_t_ = 0.058 *t* + 2.36	0.958	0.058	74.43
PCH10/HT	Visible	1/C_t_ = 1.369 *t* + 1.87	0.994	1.369	67.07
PCH15/HT	Visible	1/C_t_ = 0.874 *t* + 2.44	0.986	0.874	72.40

**Table 4 nanomaterials-11-03427-t004:** Comparison on photocatalytic activity of prepared ZnO/SiO_2_-PCH samples with other photocatalysts.

Photocatalyst	Photocatalyst Dosage (mg/L)	MV Concentration (mg/L)	Photon Source	DE at 120 min	Reference
CeO_2_/CdS/RGO	250	25	Sun light	100	[47]
KAlPO_4_F	1000	10	Visible	26	[48]
h-YbFeO_3_/o-YbFeO_3_	200	30	Visible	66.2	[49]
Nanoporous carbons	300	5 × 10^−5^	Visible	39.8	[50]
TiO_2_/Pt	2000	15	UV	95	[51]
Na_3_V_2_O_2_(PO_4_)_2_F	800	20	UV	82	[52]
Mg-doped ZrO_2_	400	5	UV	94	[53]
Activated carbon-supported NiS/CoS	5000	10	Visible	78	[54]
ZnO/SiO_2_-PCH (PCH15/HT)	800	50	UV	99.58	This work
ZnO/SiO_2_-PCH (PCH15/HT)					
ZnO/SiO_2_-PCH (PCH15/HT)	800	20	UV	100	This work

**Table 5 nanomaterials-11-03427-t005:** The change of Zn to O ratio of fresh and used PCH15/HT sample.

Sample	Binding Energy of Zn 2p_3/2_	Binding Energy of O 1s	FWHM of Zn 2p_3/2_	Atomic Ratio of Zn/O
Fresh	1030.22	531.80	2.377	0.531
After 5th cycle	1029.50	531.75	3.093	0.518

## Data Availability

The data presented in this study are available on request from the corresponding author.

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
