# Peer review of "Influencing Factors in the Synthesis of Photoactive Nanocomposites of ZnO/SiO2-Porous Heterostructures from Montmorillonite and the Study for Methyl Violet Photodegradation"

_nanomaterials, 2021, doi:10.3390/nano11123427_

Round 1

Reviewer 1 Report

Dear authors,

first of all, congratulations on their work. I list below the errors I find and the queries that arise:

Page 3 Line 114: Why were 3 days of hydrothermal treatment necessary? 

Page 4 137 lines: typo, x-ray was written in small letters x. Correct X-ray.

Page 6 fig.3: SEM image captions are incorrect: pch5/10mw and pch5/15mw. I think it's unnecessary to cram 15 SEM images together, as the characterization is no more than 10 lines. In addition to the basic montmorillonite, I would consider it sufficient, for example, to present samples containing 10 wt% Zn for microwave and hydrothermal synthesis. In addition, the element-maps that could be indicated.

Page 7 fig.4: the pressure symbol is small p, not a large P, this needs to be correct.

Page 10 fig 7 description: the figure description incorrectly uses the "b" twice, the PL spectrum diagram should already be marked "c". This same page in line 299 is also badly referred to 7b, please correct.

Page 10 fig7: why are only UV-DRS and bandgap curves of 5 w% samples indicated for samples obtained during microwave treatments?

Page 11 fig.8: in figure "a" the inscription "0" on the y-axis is not indexed. in the same figure, in both cases, the connection of measuring points impairs the transparency of the graph. I suggest you cancel it, but at least enlarge the marking of the measuring points.

Page 13 fig.10: figure "b" does not show the pch5/mw data series. I have not been clear from a description of the reasons why you should leave this. Figures "a" and "b" are slid together, partially obscuring each other's axle labels.

Page 18: from the conclusion I miss the statement that out of the 6 types examined, this was significantly different, better or worse, which synthesis or Zn ratio showed the best results, or how the absence of this can be explained. There should also be a recommendation on what direction of research the results will suggest for the future.

Overall, I miss the beginning of the thesis to give a thorough explanation of why the authors chose 5%, 10% and 15 wt%. I'm asking for a replacement.

Author Response

Reviewer#1,

  1. Page 3 Line 114: Why were 3 days of hydrothermal treatment necessary? 

Response: The reason has been added. The chosen time of treatment was referred to previous references

  1. Page 4 137 lines: typo, x-ray was written in small letters x. Correct X-ray.

Response: Thank you. The sentence has been revised

  1. Page 6 fig.3: SEM image captions are incorrect: pch5/10mw and pch5/15mw. I think it's unnecessary to cram 15 SEM images together, as the characterization is no more than 10 lines. In addition to the basic montmorillonite, I would consider it sufficient, for example, to present samples containing 10 wt% Zn for microwave and hydrothermal synthesis. In addition, the element-maps that could be indicated.

Response: the figures have been revised. The samples containing 15 wt% of Zn were chosen.

  1. Page 7 fig.4: the pressure symbol is small p, not a large P, this needs to be correct.

Response: Thank you. The symbol has been revised

  1. Page 10 fig 7 description: the figure description incorrectly uses the "b" twice, the PL spectrum diagram should already be marked "c". This same page in line 299 is also badly referred to 7b, please correct.

Response: Thank you. The graphs have been revised.

  1. Page 10 fig7: why are only UV-DRS and bandgap curves of 5 w% samples indicated for samples obtained during microwave treatments?

Response: Thank you. The graphs have been revised.

  1. Page 11 fig.8: in figure "a" the inscription "0" on the y-axis is not indexed. in the same figure, in both cases, the connection of measuring points impairs the transparency of the graph. I suggest you cancel it, but at least enlarge the marking of the measuring points.

Response: As suggested, the graph was deleted. The discussion on this has been added in the paragraph.

  1. Page 13 fig.10: figure "b" does not show the pch5/mw data series. I have not been clear from a description of the reasons why you should leave this. Figures "a" and "b" are slid together, partially obscuring each other's axle labels.

Response: Thank you. The graphs have been revised.

  1. Page 18: from the conclusion I miss the statement that out of the 6 types examined, this was significantly different, better or worse, which synthesis or Zn ratio showed the best results, or how the absence of this can be explained. There should also be a recommendation on what direction of research the results will suggest for the future.

Response:

Generally speaking, the higher Zn content gave the higher DE values as the more photoactive site cause the more reduction-oxidation occurred. In addition, hydrothermal method produced the higher specific surface area compared to the microwave-assisted synthesis, contribute to enhance the degradation via adsorption mechanism. Similar phenomenon on the role of specific surface area and external surface area for was also reported in the photocatalysis by hydrothermally synthesized-titanium pillared clay [37], tin oxide-montmorillonite [18], and zinc oxide/bentonite [38].  

  1. Overall, I miss the beginning of the thesis to give a thorough explanation of why the authors chose 5%, 10% and 15 wt%. I'm asking for a replacement.

Response: The Zn content range was set refer to many studies on supported-ZnO nanocomposite. The discussion on it has been added.

Reviewer 2 Report

In this study, Fatimah et al. prepared, characterized several photoactive nanocomposites of ZnO/SiO2-porous heterostructures from montmorillonite and evaluated their photocatalytic degradation of methyl violet (MV) under irradiation with UV and visible light. The experimental results and performances themselves seem interesting which deserve to publishing in the Nanomaterials. However, a major revision of the manuscript is required, since some issues are present:

  1. All abbreviations should be explained on their first appearance in the text, including Abstract, Highlights.
  2. In the first paragraph of the introduction section, the background studies on the photocatalytic degradation of organic compound–containing wastewater need to highlight with citations. There are some recent works could be considered as references: https://doi.org/10.1007/s11356-019-07193-5; https://doi.org/10.1016/j.jwpe.2020.101827; https://doi.org/10.1016/j.arabjc.2020.04.028; https://doi.org/10.1016/j.mtchem.2020.100380; https://doi.org/10.3390/nano11082045; https://doi.org/10.1016/j.chemosphere.2021.130163.
  3. Table 4: Kindly collect and provide the detail photon source, including light source, wavelength, intensity, etc.
  4. Figures 8, 12, 13, 14: How is the standard error for the results of figures? Could you provide the error bar for the content of figures? No replicates and/or the related appropriate statistics is reported in results and discussion of your manuscript. However, to replicate experiments and to assess the uncertainty of results is at the core of science and indispensable for reliable and high quality results. Repetition of experiments should be done.
  5. The entire text should be carefully checked for misprints and language errors.

Author Response

Reviewer#2:

  1. All abbreviations should be explained on their first appearance in the text, including Abstract, Highlights.

Response: Thank you. The abbreviations in abstract and in throughout manuscript have been revised.

  1. In the first paragraph of the introduction section, the background studies on the photocatalytic degradation of organic compound–containing wastewater need to highlight with citations. There are some recent works could be considered as references: https://doi.org/10.1007/s11356-019-07193-5; https://doi.org/10.1016/j.jwpe.2020.101827; https://doi.org/10.1016/j.arabjc.2020.04.028; https://doi.org/10.1016/j.mtchem.2020.100380; https://doi.org/10.3390/nano11082045; https://doi.org/10.1016/j.chemosphere.2021.130163.

Response: Some references have been added.

  1. Table 4: Kindly collect and provide the detail photon source, including light source, wavelength, intensity, etc.

Response: the information on detail of photon source and intensity have been added.

  1. Figures 8, 12, 13, 14: How is the standard error for the results of figures? Could you provide the error bar for the content of figures? No replicates and/or the related appropriate statistics is reported in results and discussion of your manuscript. However, to replicate experiments and to assess the uncertainty of results is at the core of science and indispensable for reliable and high quality results. Repetition of experiments should be done.

Response: The standard errors have been added

  1. The entire text should be carefully checked for misprints and language errors.

Response: Thank you. The manuscript has been proof-read and checked for errors.

Round 2

Reviewer 2 Report

Accept in present form.